# Competing effects of vegetation density on sedimentation in deltaic marshes

Yuan Xu [1,2] ✉, Christopher R. Esposito [3,4], Maricel Beltrán-Burgos[4] & Heidi M. Nepf[2]

Marsh vegetation, a definitive component of delta ecosystems, has a strong effect on sediment retention and land-building, controlling both how much sediment can be delivered to and how much is retained by the marsh. An understanding of how vegetation influences these processes would improve the restoration and management of marshes. We use a random displacement model to simulate sediment transport, deposition, and resuspension within a marsh. As vegetation density increases, velocity declines, which reduces sediment supply to the marsh, but also reduces resuspension, which enhances sediment retention within the marsh. The competing trends of supply and retention produce a nonlinear relationship between sedimentation and vegetation density, such that an intermediate density yields the maximum sedimentation. Two patterns of sedimentation spatial distribution emerge in the simulation, and the exponential distribution only occurs when resuspension is absent. With resuspension, sediment is delivered farther into the marsh and in a uniform distribution. The model was validated with field observations of sedimentation response to seasonal variation in vegetation density observed in a marsh within the Mississippi River Delta.

Natural and anthropogenic forces, such as sea-level rise[1,2], coastal development[3], reduction of riverine sediment[4,5], and subsidence and compaction of coastal sediments[6,7], have caused extensive land loss, degradation, and fragmentation of coastal ecosystems, threatening the delivery of important ecosystem services[8]. For example, the average relative sea-level rise in the Mississippi River Delta (MRD) is 13 mm/yr[9]. Over the past century, rapid wetland loss has occurred in coastal Louisiana, with a total area of ≈5000 km² lost between 1932 and 2016[10]. A major restoration scheme is planned, including two sediment diversions, which will enhance sediment delivery and simulate the natural delta-building processes[4,11]. However, it is still unclear how much sediment can be delivered to and retained by the coastal wetlands[2,12].

Wetland vegetation has a strong influence on sediment transport in river deltas[13,14]. It is widely believed that vegetation enhances sedimentation in both fresh- and saltwater marshes as vegetation increases hydraulic roughness, which reduces flow velocity[15], enhancing sediment retention, stabilizing deposited sediment, and minimizing erosion[16,17] However, some field observations and numerical models have shown that vegetation does not always enhance sedimentation on deltaic marshes[13,18]. Because flow is diverted away from regions of very dense vegetation, the sediment supply to those regions declines, reducing the potential deposition[19,20]. Nardin and Edmonds[13] first described these competing influences of vegetation, enhanced retention, and reduced supply, showing that both can impact the net accumulation of sediment on a marsh platform. At present, there are few studies that consider both processes or quantify their competing effects on sedimentation and its spatial distribution. In addition, most model studies, e.g., refs. 13, 21, simplify the vegetation effects through a Manning's roughness, which does not account for the additional turbulence generated by vegetation, which can also impact sediment transport[22–24].

[1]State Key Laboratory of Hydroscience and Engineering, Department of Hydraulic Engineering, Tsinghua University, Beijing, China. [2]Department of Civil and Environmental Engineering, Massachusetts Institute of Technology, Cambridge, MA, USA. [3]The Water Institute of The Gulf, Baton Rouge, LA, USA. [4]Department of Earth and Environmental Sciences, Tulane University, New Orleans, LA, USA. ✉e-mail: xuyuan18@mails.tsinghua.edu.cn

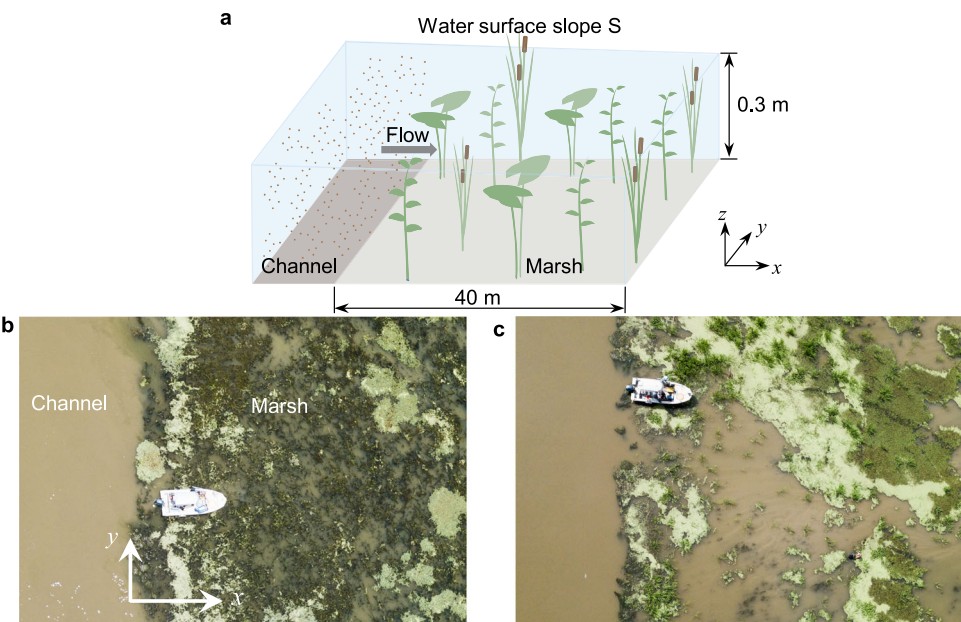

**Fig. 1 | Modeled scenario including a channel and a marsh with emergent vegetation. a** Schematic of modeled scenario with typical scales. **b** Photo of a field site in Cubits Gap sub-delta, MRD, taken in June 2019, corresponding to the highest vegetation coverage, $V_c = 66\%$. **c** Same as **b**, but in July 2019 with $V_c = 20\%$. Change in vegetation coverage was due to natural variation over the growing season. The dominant vegetation was *Potamogeton nodosus*.

Understanding the influence of vegetation on the competing mechanisms of sediment supply and retention is critical for planning successful restoration strategies that will recover lost land. For example, a more efficient application of sediment diversion could be achieved by timing sediment input to periods of vegetation density that optimize sediment accretion, which was the focus of this study. Specifically, this study considered the impact of vegetation density on the sediment supply to and retention on a marsh platform using a random displacement model (RDM), a Lagrangian method that tracks the transport of sediment particles subject to advection, diffusion, deposition, and entrainment. The RDM method was chosen for its computational efficiency. Unlike most numerical models, which represent vegetation through a Manning's roughness[13,20,21], our model considered the effects of vegetation size and area density on velocity, turbulence, and diffusivity, each of which influences sediment deposition and resuspension. Rather than considering a specific site, this study intentionally considered a reduced-order (2D) representation of a marsh platform in order to explore a wide parameter space. The RDM method enabled us to separately quantify the effects of supply and retention and their impacts on sedimentation and its spatial distribution. A previous study validated the RDM model using laboratory measurements of suspended sediment concentration[22]. The present study offers further validation through a comparison to field measurements.

The model scenario is based on a deltaic marsh and focuses on the exchange between a channel and a marsh platform with emergent vegetation (Fig. 1a). The flow entering the marsh is steady, perpendicular to the marsh edge, and driven by a water surface slope between the channel and marsh. Sediment is continuously introduced to the marsh at the channel boundary. Based on typical conditions in marshes[18–21], 16 base cases considered vegetation area density $n = 0$ to 500 stems/m², with stem diameter $d = 1.0$ cm, water depth $H = 0.3$ m, and water surface slope $S = 0.0005$. To consider seasonal effects on vegetation growth and water flux[14], additional slopes ($S = 0.00025$ and 0.001) and stem diameters ($d = 0.5$ and 1.5 cm) were considered, for a total of 80 cases. Finally, the model was compared to a field study within the MRD, reported in ref. 19, which measured flow, sediment, and vegetation parameters in different seasons (April 24, June 3, and July 1, see also Fig. 1b, c).

## Results

### Effects of vegetation density on flow conditions

The impacts of vegetation on flow and sediment fate can be quantified by three parameters. First, vegetation increases flow resistance, which decreases velocity on the marsh, which in turn reduces sediment supply relative to the bare bed reference state. The restriction of supply is quantified by the sediment supply ratio (SSR),

$$\text{SSR} = \frac{M_{in,n}}{M_{in,0}} = \frac{C_c H U_n}{C_c H U_0} = \frac{U_n}{U_0} \tag{1}$$

$M_{in,n}$ is the sediment mass flux entering the marsh carried by fluid velocity $U_n$, associated with vegetation density $n$ denoted with subscript "$n$", and $n = 0$ for a bare bed (without vegetation). $C_c$ is the suspended sediment concentration in the channel.

Second, sediment retention is quantified by the ratio of net deposition to sediment supply, called the retention efficiency (RE), with $M_{nd,n}$ the total net deposition on the marsh with vegetation density $n$.

$$\text{RE} = \frac{M_{nd,n}}{M_{in,n}} \tag{2}$$

Third, to compare between cases, the total net deposition, $M_{nd,n}$, was normalized by the mass supplied to the bare marsh platform (i.e., maximum supply, $M_{in,0}$), and this ratio is called the relative net deposition (ND)

$$\text{ND} = \frac{M_{nd,n}}{M_{in,0}} \tag{3}$$

Vegetation density impacts time-mean and turbulent velocity, which influences sediment supply and resuspension. Specifically, velocity, shear stress, turbulent kinetic energy (TKE), and diffusivity all decrease with increasing vegetation density (Fig. 2). For a fixed slope $S$ and stem diameter $d$, increasing stem density $n$, which increases flow resistance, reduces velocity[25,26]. A reduction in velocity is associated with a reduction in sediment supply. Because flow resistance is

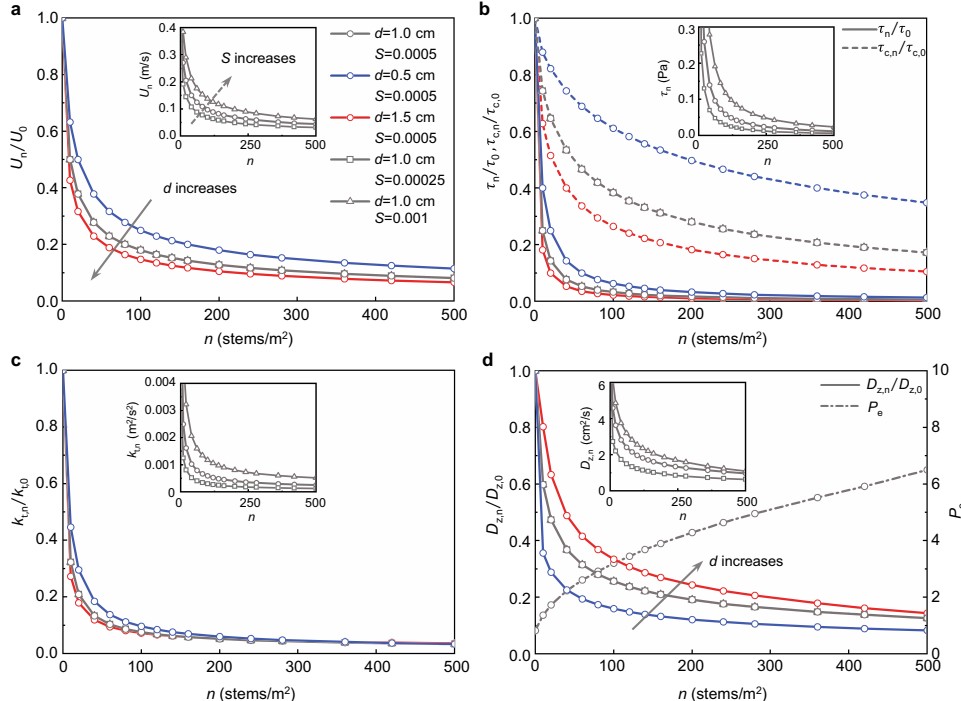

**Fig. 2 | Changes in flow parameters with increasing vegetation area density, $n$.** Each parameter is normalized by its value without vegetation (bare bed), denoted by subscript "0". **a** Normalized velocity ($U_n/U_0$) decreases with increasing stem density $n$. Blue to red color represents increasing diameter or slope. The normalized velocity is not a function of slope (Eq. (M4)), so all cases with the same stem diameter but different slopes collapse to a single curve, shown by the gray curve. The inset graph shows the dimensional velocity $U_n$ for different slopes with $d = 1.0$ cm. **b** Same as **a**, but for normalized bed shear stress ($\tau_n/\tau_0$, solid lines) and critical shear stress ($\tau_{c,n}/\tau_{c,0}$, dashed lines). The inset graph shows the variation in shear stress for different slopes with $d = 1.0$ cm. The critical shear stress is not a function of the slope. **c** Same as **a**, but for normalized turbulent kinetic energy ($k_{t,n}/k_{t,0}$). **d** Same as **a**, but for normalized diffusivity ($D_{z,n}/D_{z,0}$). The dash-dot line indicates the Peclet number ($P_e$), the right-hand axis.

proportional to the frontal area, for the same stem density, a larger (smaller) stem diameter produces a smaller (larger) velocity (Fig. 2a). Increasing slope increases the velocity (Fig. 2a, inset graph), but does not influence the velocity ratio (Eq. 1), so that supply restriction is not a function of surface slope (Eq. (M4) in Methods). Because bed shear stress is proportional to velocity squared ($\tau \sim U^2$, Eq. (M9)), it declines more rapidly than velocity with increasing stem density (Fig. 2b). The significant decrease in $\tau_n$ with increasing $n$ reduces the tendency for sediment resuspension, which enhances retention.

Previous studies have shown that for the same velocity, turbulent kinetic energy, $k_t$, is higher in vegetated flow than in non-vegetated flow due to vegetation-generated turbulence[27] (Methods, Eq. (M5)). However, since velocity decreases with increasing $n$, which also impacts TKE, the TKE decreases with increasing stem density (Fig. 2c). For the same $n$, increasing $d$ (Fig. 2c, blue to red) reduces velocity, which tends to reduce TKE. However, increasing $d$ also increases solid volume fraction, $\phi$, which tends to enhance TKE[15]. These competing effects result in only a small variation in TKE even with a large variation in $d$.

Turbulent diffusivity ($D_z$) is proportional to the turbulent velocity scale $\sqrt{k_t}$ (Methods), such that diffusivity decreases with increasing stem density (Fig. 2d). Turbulent diffusion tends to carry sediment away from the bed, inhibiting deposition, so that a decrease in diffusivity facilitates sedimentation. The Peclet number $P_e = w_s H/D_z$ (right axis in Fig. 2d), which measures the relative importance of settling and diffusivity[28], increases as stem density $n$ increases, indicating a shift in sediment transport dominated by diffusion to transport dominated by the settling. This shift impacts the shape of the suspended sediment concentration (SSC) profile. Specifically, as $P_e$ increases with $n$, the vertical profile of SSC becomes increasingly non-uniform, with a higher concentration near the bed (Supplementary, S2).

Finally, because vegetation-generated turbulence can contribute to resuspension, the critical shear stress for resuspension[29,30], $\tau_c$, is lower with vegetation than for bare bed, and it decreases with increasing stem density (Fig. 2b). Further, for the same stem density, it decreases with increasing stem diameter (Methods, Eqs. (M8) and (M10)). Importantly, bed stress ($\tau_n$) decreases more rapidly with increasing stem density than critical shear stress $\tau_{c,n}$, so the overall effect of increasing vegetation density is to reduce resuspension.

## Effects of vegetation on sedimentation

Increasing stem density $n$ reduces velocity (Fig. 2a), which reduces sediment supply to the marsh, illustrated by the supply ratio, SSR (Eq. (1), Fig. 3a). However, a reduction in velocity is also associated with a reduction in shear stress, TKE, and diffusivity, and each of these trends favors deposition, increasing sediment retention efficiency RE (Eq. (2), Fig. 3a). The competing trends of SSR and RE produce a nonlinear relationship between net deposition (ND = SSR × RE) and vegetation density, such that an intermediate density yields the maximum net deposition (Fig. 3b). This is an important result, because one often thinks of vegetation only in terms of enhancing deposition, but the degree of enhancement has a strong dependence on vegetation density through the influence on sediment supply. For example, in the base cases (Fig. 3b), the optimal density ($n = 80$ m$^{-2}$) traps 16% of the bare bed supply, whereas the highest density ($n = 500$ m$^{-2}$) only traps 7.6% of $M_{in,0}$, i.e., the densest vegetation does not trap the most sediment. For high vegetation density $\tau_n < \tau_{c,n}$ ($n \geq 160$ m$^{-2}$ in Fig. 3b), which eliminates resuspension, making retention efficiency RE $\approx 1$, so that sediment accretion depends only on supply restriction. In contrast, for small stem density ($n < 160$ m$^{-2}$ in Fig. 3b), $\tau_n > \tau_{c,n}$, and resuspension is active and increases with decreasing $n$, offsetting the greater supply (SSR) with

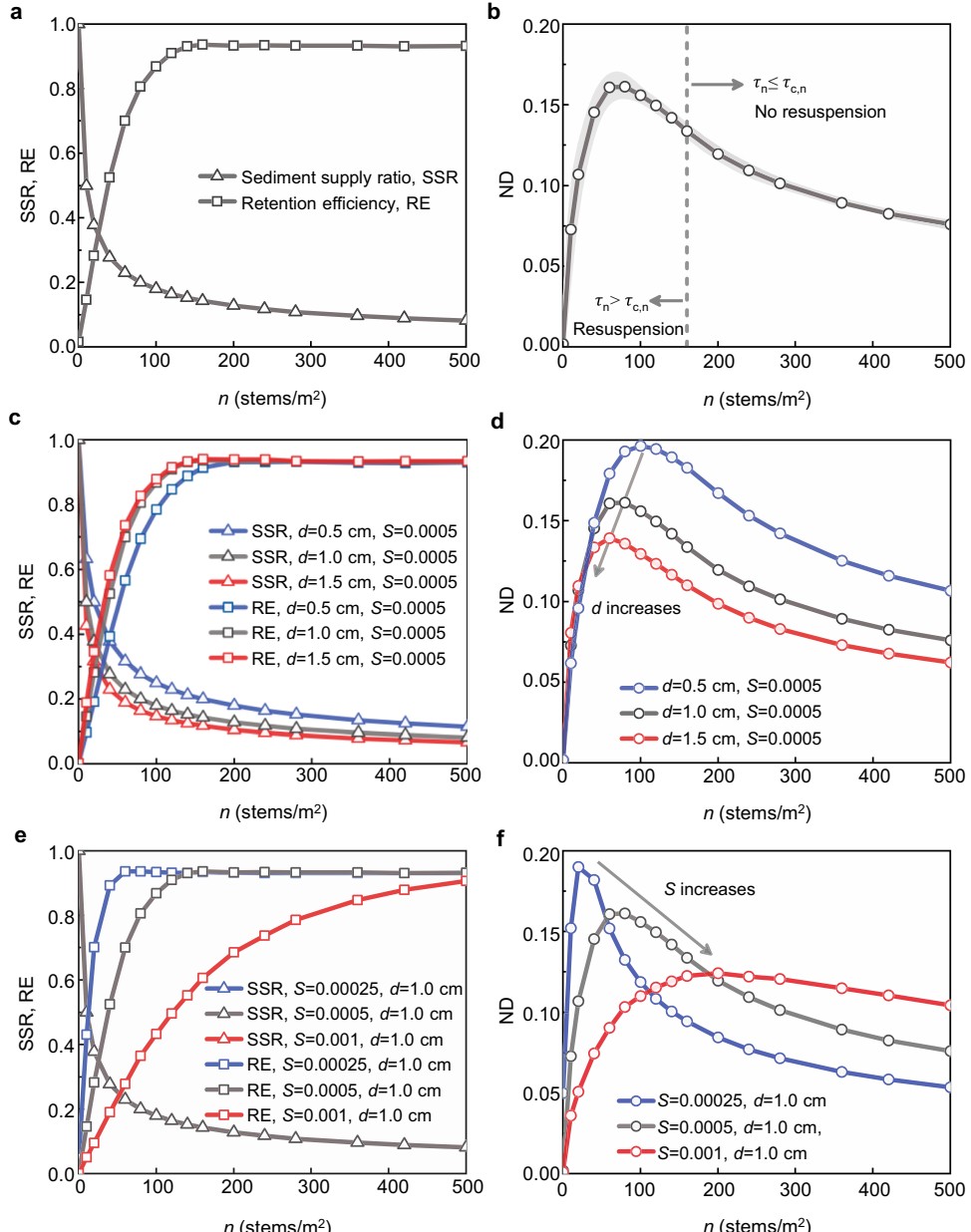

**Fig. 3 | Modeled sedimentation (net deposition) with vegetation area density *n*.** SSR is sediment supply ratio, triangles (Eq.1). RE is retention efficiency, squares (Eq. 2). ND is normalized net deposition, circles (Eq.3). Base cases in **a** and **b**, varied stem diameter *d* in **c** and **d**, and varied slope *S* in **e** and **f**. Blue to gray to red color corresponds to increasing diameter or slope. In **b** vertical dashed line at $n = 160$ m$^{-2}$ indicates the condition $\tau_n = \tau_{c,n}$. The gray shading around the ND curve denotes the ensemble standard deviation based on ten realizations (Methods), which ranged from 1.1 to 6.0% of the ensemble average. In **e** note that SSR is not a function of slope (Eqs. (1) and (M4)), so SSR curves with different slopes overlap, shown by the gray line with triangles.

decreasing *n*, so that net deposition (ND) decreases as *n* approaches 0.

Differences in vegetation type and growth phase can be represented through stem diameter *d*. Specifically, increasing *d* indicates a greater frontal area per plant stem, which increases flow resistance and decreases velocity (Fig. 2a), so that sediment supply, SSR, is diminished with increasing *d* (Fig. 3c). This trend dominates the net deposition over most of the stem density range, such that ND decreases with increasing *d* (Fig. 3d). However, for small *n*, and specifically when resuspension is active ($\tau_n > \tau_{c,n}$), increasing *d* decreases bed shear stress by a greater degree than it decreases the critical shear stress (Fig. 2b), such that resuspension decreases with increasing *d*, which enhances the retention efficiency (RE in Fig. 3c). For this reason, when *n* is small ($n < 40$ m$^{-2}$), net deposition increases with increasing *d* (Fig. 3d).

Variation in flow forcing due to seasonality, storms, or tides, can be represented by varying *S*. Consider the smallest value, $S = 0.00025$, for which the densest vegetation yields the same net deposition as an unvegetated marsh (ND ≈ 5% for both $n = 0$ and 500 m$^{-2}$, Fig. 3f). That is, the vegetation provides no enhancement to sediment accretion. This is representative of lower energy environments ($S < 0.00025$), for which $\tau_0 \leq \tau_{c,0}$ even for the bare bed, such that the peak in sedimentation shifts to $n = 0$, and the net deposition in all vegetation cases will be smaller than the unvegetated case. This is because in the absence of resuspension, net deposition is controlled only by supply, which decreases with increasing vegetation density. As *S* increases from 0.00025 to 0.001, the peak in net sedimentation shifts from $n = 20$ to $n = 200$ m$^{-2}$, and the magnitude decreases (Fig. 3f). These shifts occur because increasing *S* increases velocity and thus $\tau_n$, which

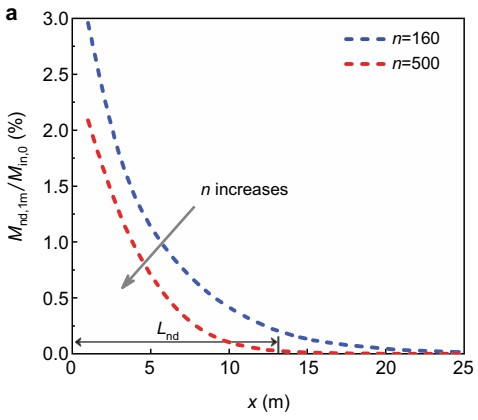
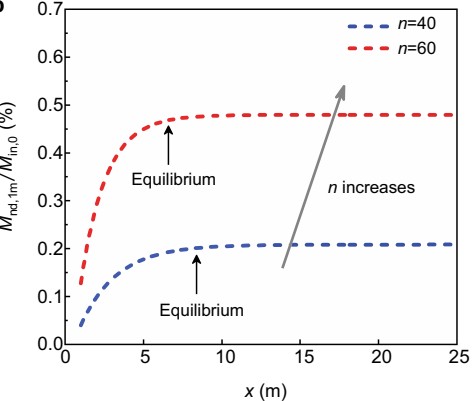

**Fig. 4 | Two spatial patterns of sedimentation. a** Pattern 1. In cases without resuspension ($\tau_n < \tau_{c,n}$ e.g., base cases $n = 160$ and $500$ m$^{-2}$) normalized sedimentation $M_{nd,1m}/M_{in,0}$, estimated over $\Delta x = 1$ m increments, decreases with distance from marsh edge over deposition length-scale $L_{nd}$, the maximum distance sediment can be carried into the marsh. The curves of $M_{nd,1m}$ (dashed lines) were smoothed with a moving average. **b** Pattern 2. In cases with resuspension ($\tau_n > \tau_{c,n}$, e.g., base cases $n = 40$ and $60$ m$^{-2}$) $M_{nd,1m}/M_{in,n}$ increases with distance from marsh edge until SSC profile reaches equilibrium at $L_{eq}$, indicated by black arrows.

enhances resuspension and reduces retention efficiency for small $n$ (Fig. 3e, f). Consequently, the peak progressively shifts to a larger $n$ but has a smaller magnitude.

### Effects of vegetation on sedimentation spatial distribution
The vegetation density can also affect the spatial distribution of sedimentation[31,32]. Two deposition patterns emerge in the model, depending on whether resuspension is present. If resuspension is absent ($\tau_n < \tau_{c,n}$), deposition progressively decreases with distance into the marsh ($x$ direction), called Pattern 1 (Fig. 4a). Pattern 1 is common in freshwater marshes[32,33]. Maximum deposition occurs at the marsh edge, where supply is highest. Since no resuspension occurs, deposition near the edge reduces the supply to regions farther into the marsh, and the progressive decrease in supply is reflected in the progressive decrease in sedimentation with distance from the marsh edge. This distribution has been described by an exponential model for sediment deposition[31]. Because the velocity (Fig. 2) and supply (Fig. 3) decrease with increasing stem density, in Pattern 1, both the total sedimentation (area under curve, dependent on supply) and the distance over which deposition occurs, $L_{nd}$ (dependent on velocity) decrease with increasing $n$ (Fig. 4a). While total deposition increases with time, the sedimentation distance does not change. Specifically, for Pattern 1, sediment distribution is dominated by settling (large $P_e$), so that the settling time-scale, $H/w_s$, defines the sedimentation distance, i.e., $L_{nd} = (2.0 \pm 0.1)\frac{UH}{w_s}$ (Supplementary, S3).

When resuspension is present (Pattern 2), particles deposited near the edge can later be resuspended and delivered farther into the marsh so that sedimentation can occur farther into the marsh than in Pattern 1 (Fig. 4b). In Pattern 2, sedimentation increases with distance from the marsh edge until distance $L_{eq}$, reflecting the adjustment in the vertical profile of SSC. At the marsh edge, sediment is introduced uniformly over the depth, which simulates well-mixed sediment coming from the river. As suspended sediment travels into the marsh, the SSC profile evolves to an equilibrium distribution with higher SSC near the bed (Supplementary, Fig. S4). As a result, the near-bed SSC increases between the marsh edge and marsh interior, which results in the increase in sedimentation (Fig. 4b). The distance over which this adjustment occurs, $L_{eq}$, decreases with increasing stem density (Fig. 4b). $L_{eq}$ is set by the adaptation time-scale for the SSC profile. For small $P_e$ characteristic of Pattern 2, this is the diffusion time-scale, $T_a \sim H^2/D_z$[25,31], and $L_{eq} \sim U T_a$. Specifically, $L_{eq} = (0.16 \pm 0.02)\frac{UH^2}{D_z}$ (Supplementary, S3).

### Validation with field measurements
The simulation of net deposition was validated by comparison to a field study reported in Beltrán-Burgos[19], which measured velocity, sediment, and vegetation parameters over a growing season at Cubits Gap, a sub-delta in the MRD (Fig. 1b, c). At this site, the maximum sedimentation occurred during a period of intermediate vegetation density (July). The model inputs, based on the field measurements, are summarized in Table 1 and described in the Methods. The measured (stars) and modeled (circles) sedimentation rates, $\widetilde{q_d}$, had a good agreement (Fig. 5), indicating that the RDM captured the competing effects of supply restriction and retention efficiency to predict the observed nonlinear dependence between vegetation density and net deposition.

## Discussion
Using a reduced-order representation of flow between a channel and marsh platform, this study explored a wide parameter space, which provided insight into the role of vegetation density on sediment accretion. First, as vegetation density increases, velocity decreases, which is associated with a reduction in shear stress, TKE, and diffusivity, all of which favor deposition and sediment retention. However, the reduction in velocity also reduces sediment supply. The competing effects of enhanced retention and reduced supply produce a nonlinear relationship, such that an intermediate vegetation density yields the maximum net deposition. Second, the optimum vegetation density depends on the water surface slope ($S$), with maximum accretion shifting toward lower vegetation density as $S$ decreases. In particular, for sufficiently small $S$, such that $\tau_0 \le \tau_{c,0}$ even for the bare bed, the

### Table 1 | Field data in Cubits Gap sub-delta

|          | H (m) | U (cm/s) | $V_c$ (%) | LAI (m²/m²) | $n_l$ (leaves/m²) | S       | $q_d$ (g/m²/day) |
|----------|-------|----------|-----------|-------------|-------------------|---------|------------------|
| April 24 | 0.4   | 13       | 5         | 0.05        | 16                | 0.00013 | 14               |
| June 3   | 0.6   | 3        | 66        | 1.02        | 340               | 0.00008 | 190              |
| July 1   | 0.6   | 6        | 20        | 0.21        | 70                | 0.00007 | 324              |

Velocity, $U$, from drone images of tracer motion, and vegetation coverage, $V_c$, measured on dates shown in the table. Leaf area index, LAI (one-sided leaf area per bed area) estimated from $V_c$ using method described in ref. 63 (See Methods). The leaves per bed area, $n_l$ = LAI/$A_l$, with $A_l$ the one-sided area of one leaf. The water surface slope, $S$, was not measured in the field study, but was estimated from Eq. (M4) and the measured velocity, using $ndH$ = LAI. Sedimentation rate, $q_d$, for each date was estimated from net deposition recorded on tiles just before and just after the vegetation and velocity measurement dates (see Fig. 41b mudflat site in Beltrán-Burgos[19]). The grain size in the field was between 12 and 53 µm, and in the model the size was set to 50 µm.

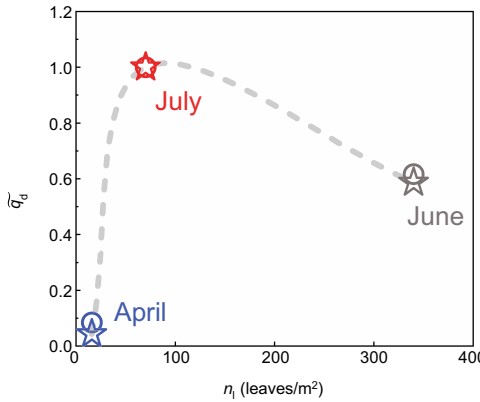

**Fig. 5 | Comparison between measured (stars) and simulated (circles) sedimentation rate.** The gray dashed curve was added to emphasize the nonlinear relationship between net deposition rate and vegetation density. Note that each measurement occurred under a different water slope (Table 1), which was associated with a different modeled curve of $\widetilde{q_d}$ versus vegetation density. The individual curves are shown in Supplementary S5.

peak accretion is associated with bare bed ($n = 0$), and sediment accretion rate decreases with the addition of vegetation. Finally, vegetation density also affects the spatial distribution of sedimentation. If vegetation density is sufficient to eliminate resuspension, the deposition pattern is exponential, with maximum deposition at the marsh edge (Pattern 1). However, if the vegetation density is not sufficient to eliminate resuspension, particles deposited near the edge can be resuspended and delivered farther into the marsh, resulting in lower accretion at the marsh edge and a more uniform spatial pattern of deposition farther into the marsh (Pattern 2).

Topography observed in the field suggests that sedimentation patterns can resemble a hybrid of Patterns 1 and 2 (Fig. 4), with maximum sedimentation occurring 10 s of meters from the edge (as in Pattern 2), but following Pattern 1 farther from the edge[34]. Elevated turbulence at the channel edge, which was not included in the simulation, can keep particles vertically well-mixed and promote resuspension, both of which diminish net deposition near the edge. The penetration of turbulence from the channel, as well as local shear-layer turbulence and edge-wave generation, could impact the marsh over distance $\delta \sim (nd)^{-1}$ [25,35]. However, this length-scale is O(1 m), which is small compared to the equilibrium distance, $L_{eq}$, suggesting $L_{eq}$ is the more important length-scale for edge morphology. In the field, levee elevation increases over a greater length-scale, O(10 m), suggesting a larger transition length for deposition. Seasonal, storm-related, and tidal variation in water depth that submerges the marsh vegetation may extend the deposition length, as higher velocity above the vegetation can carry sediment farther into the marsh, a process not captured in our model, which only considered emergent vegetation.

An important caveat in the simulation is that all deposited particles have the same opportunity to resuspend, dictated by a constant $\tau_c$. In real systems, this could describe newly deposited sediment or deposition within a specific flow event but would likely overestimate the resuspension of particles deeper in the sediment profile because $\tau_c$ typically increases with sedimentation depth because of consolidation and physico-chemical effects[36]. Thus, while the spatial pattern may be correctly captured, the simulation underestimates the long-term accumulation of sediment. Moreover, the over-estimation of the resuspension rate may shift the optimal vegetation density upward, suggesting a smaller optimum $n$ in real systems relative to the RDM results.

The 2D simulation (streamwise-depth plane) cannot resolve horizontal heterogeneity in vegetation density (patchiness), which can be

present in the field (e.g., Fig. 1c). When vegetation area coverage is above a percolation threshold (>40–50% area coverage), patchiness does not significantly impact marsh-scale flow, and a spatial average model, such as the present simulation, is accurate[37,38]. Below this threshold, the effect of patchiness depends on stem density. For sparse vegetation ($C_d ndH \lesssim 0.5$, i.e., $n < 160 \, \text{m}^{-2}$ in our study), patchiness has a negligible impact on flux[38]. For dense vegetation ($C_d ndH \gtrsim 1.0$, i.e., $n > 330 \, \text{m}^{-2}$ in our study), patchiness enhances channel-marsh connectivity, which increases sediment supply but also increases local velocity and shear stress, which can increase resuspension and decrease retention. Because both supply and resuspension are enhanced, the impact of patchiness on sediment accretion may be small.

By considering the interaction between flow, vegetation, and sediment, this study provides important insight into how vegetation density (related to species and growth phase) influences the potential to trap sediment and build marsh, which are key processes needed to save drowning coasts, such as Louisiana, USA. In particular, the study can provide guidance to optimize the retention of sediment by wetlands targeted by sediment diversions, both at the scale of local levee cuts and by the billion-dollar Mid-Barataria and Mid-Breton structures.

## Methods
### Modeling method
A random displacement model (RDM) is a Lagrangian method that tracks the movement of individual sediment particles, including advection, diffusion, settling, and entrainment (resuspension). It is derived from the Fokker–Planck equation, which describes the conditional probability for the particle's velocity and position as a function of time[39]. For a large number of particles and a small time-step, the Fokker–Planck equation is equivalent to the continuum advection-diffusion equation[40,41]. In this study, a 2D (streamwise, $x$, and vertical, $z$, Fig. 1a) RDM was used to simulate the transport of individual sediment particles over a marsh platform. Sediment particles uniformly distributed over depth were continuously introduced to the marsh at the channel boundary. Within each time-step, $\Delta t$, the displacement of each particle ($\Delta x$, $\Delta z$) is derived from two components: the time-mean advection and a random turbulent velocity[42]. Assuming a high streamwise Peclet number, the longitudinal diffusion was neglected[43]. Specifically, the particle position at time-step $i + 1$ was described as follows[22,44]:

$$x_{i+1} = x_i + u(z_i) \triangle t, \tag{M1}$$

$$z_{i+1} = z_i + \left( \frac{\partial K_z}{\partial z}(z_i) - w_s \right) \triangle t + G \sqrt{2 K_z(z_i) \triangle t}, \tag{M2}$$

in which $K_z$ is the sediment vertical diffusivity. The term $\partial K_z / \partial z$ is a pseudo-velocity needed to prevent the artificial accumulation of particles in regions of low diffusivity[45]. $G$ is a random variable with standard Gaussian distribution (zero mean and unit variance). In this study, particles in the marsh were fine enough to assume that the sediment diffusivity $K_z$ was equal to turbulent diffusivity $D_z$[46]. It is worth noting that the 2D model is valid along any vector parallel to the flow direction entering the marsh, as long as the transport onto the marsh platform is dominated by advection. Situations in which the exchange with the marsh is predominantly by lateral diffusion would not be represented by this model. For simplicity, in this study, the flow direction is assumed to be perpendicular to the marsh edge (Fig. 1), and lateral diffusion is not considered.

The time-step, $\Delta t$, was chosen so that the vertical particle trajectory within each time-step was much smaller than the scale of vertical gradients in the diffusivity and velocity[47,48]. A length-scale

equal to $0.05H$ was suggested for emergent vegetation[49]. Thus, $\Delta t$ was selected as

$$\triangle t < \min\left(\frac{0.05H}{\left|\frac{\partial D_z}{\partial z} - w_s\right|_{max}}, \frac{(0.05H)^2}{(D_z)_{max}}\right). \tag{M3}$$

The velocity, shear stress, TKE, and turbulent diffusivity were dependent on vegetation area density, $n$. For simplicity, the vegetation was modeled as rigid cylinders with diameter $d$ and height $h$ greater than water depth $H$, i.e., emergent vegetation. The velocity ($U$) within the marsh can be derived from the conservation of momentum[20,50],

$$U = \sqrt{\frac{gHS}{C_f + 0.5C_d ndH}}, \tag{M4}$$

in which $g$ is the acceleration of gravity, $C_f$ is the bed-drag coefficient, and $C_d$ is a bulk vegetation drag coefficient. The turbulent diffusivity, $D_z$, depends on the turbulence velocity scale ($k_t^{-1/2}$) and integral length-scale ($l_t$), i.e., $D_z = \alpha_z \sqrt{k_t} l_t$, in which $\alpha_z$ is a scale constant that depends on both the solid volume fraction, $\phi$, and vegetation arrangement[51]. For the marsh considered in this study, $\alpha_z = 0.44$ to $0.93$. Within vegetation, $l_t$ and $k_t$ are shaped by the vegetation. When $d \leq \Delta s$ (average distance to nearest neighboring stem), $l_t = d$[15,51]. The vegetation-generated $k_t$ within a marsh of rigid, emergent stems can be estimated from Tanino and Nepf[51], $k_t = \gamma^2 \left(C_d \frac{nd^2}{2(1-\phi)}\right)^{2/3} U^2$, in which $\gamma$ is a scale coefficient ($\gamma^2 = 1.1 \pm 0.2$) and $\phi = \pi nd^2/4$ is the solid volume fraction. The bed-generated turbulence is related to the bed shear stress ($\tau = \rho C_f U^2$). Specifically, $\tau/\rho = \xi k_t$ with the scale coefficient $\xi = 0.20 \pm 0.01$[52]. Combining the vegetation- and bed-generated turbulence, the total turbulence within the marsh is

$$k_t = \underbrace{\frac{C_f}{\xi}U^2}_{bed} + \underbrace{\gamma^2 \left(C_d \frac{nd^2}{2(1-\phi)}\right)^{2/3} U^2}_{vegetation} \tag{M5}$$

A no-flux (reflecting) boundary condition was applied to the water surface. Unlike many previous studies, which define the bed as a reflecting condition[22,49], our model accounted for sediment deposition and resuspension at the bed. Within the model, the vegetation impacts sedimentation through its influence on diffusivity and resuspension. Particles are deposited on the bed if the vertical displacement within a time-step passed below the bed level (i.e., $z$ position at time-step $i$, $z_i \leq 0$). If a particle is deposited, it will cease to move until it is re-entrained. A probabilistic approach was used to represent resuspension[53]. The resuspension probability was determined by the frequency of entrainment of a particle from the bed, $f_e$, described by the erosion rate, $E$ (m/s). Specifically, the frequency $f_e = E/D$, in which $D$ is the sediment grain size. For each deposited particle, a random number $G_r$ from a uniform distribution (0,1) was generated at each time-step $\Delta t$. If $G_r \leq f_e \Delta t$, the deposited particle resuspended[53], otherwise it remained deposited.

**Erosion of sediment and critical shear stress within the marsh**
A nonlinear erosion model provides a good description of the erosion of cohesive sediment[54,55]. There is no universally accepted methodology to estimate the critical shear stress $\tau_c$ from soil properties[56], so the best method is a direct measurement in the laboratory or field. Following the method in Walder[55], measurements of erosion rate $E$ (m/s) and bed shear stress $\tau$ in the Lower Mississippi River Delta, Barataria and Breton Sound (unpublished data, The Water Institute of the Gulf)

were fit to the following model,

$$\widetilde{\Phi} = \beta \widetilde{\tau}_e^{\ m} \tag{M6}$$

in which $\widetilde{\Phi}$ is the dimensionless erosion rate, $\widetilde{\Phi} = (\rho_d/\rho_s)\left(E/\sqrt{\frac{\tau_c}{\rho}}\right)$ with $\tau_c$ the critical shear stress, $\rho$ the fluid density, $\rho_s$ the grain density, and $\rho_d$ the bulk dry density; $\widetilde{\tau}_e$ is the dimensionless excess shear stress, $\widetilde{\tau}_e = (\tau - \tau_c)/\tau_c$. The data provided a fitted estimate of scale coefficient $\beta$; exponent $m$, and critical shear stress $\tau_c$. Thus, the erosion rate can be estimated by

$$E = 0.00034 \left(\frac{\rho_s}{\rho_d}\right)^{0.5} \left(\frac{\tau_c}{\rho}\right)^{0.5} \left(\frac{\tau - \tau_c}{\tau_c}\right)^{1.01}, \tag{M7}$$

When $\tau \leq \tau_c$, the erosion rate is zero, i.e., no resuspension. Note that $\tau_c$ was used in fitting the erosion function was determined with Sedflume under bare bed conditions[57].

Recent studies have shown that turbulence generated by vegetation can lower the threshold for sediment motion, reducing the critical shear stress $\tau_{c,n}$ and critical velocity $U_{c,n}$ within the marsh[24,29,30,58]. For flat bare beds, the role of turbulence is inherently represented in $\tau_c$ because bed shear stress has a linear relationship with TKE ($\tau/\rho = \xi k_t$)[59]. With this relation, $\tau_c$ can be converted to an equivalent turbulence threshold[58]. If we assume that the magnitude of TKE determines the onset of resuspension, we can equate the critical turbulence defined for bare bed, $k_{t,c0}$ ($=\tau_c/\xi\rho$, with subscript "0" for bare bed), to the TKE in a marsh of stem density $n$, and solve for the ratio of critical velocity for resuspension in vegetated ($U_{c,n}$) and bare ($U_{c,0}$) beds, as in Yang[58]:

$$\frac{U_{c,n}}{U_{c,0}} = \frac{1}{\sqrt{1 + \frac{\xi\gamma^2}{C_f}\left(\frac{C_d}{2}\frac{nd^2}{1-\phi}\right)^{2/3}}} \tag{M8}$$

For consistency with previous work, we cast this in terms of the Shields parameter,

$$\theta = \frac{\tau}{(\rho_s - \rho)gd_{50}} = \frac{\rho C_f U^2}{(\rho_s - \rho)gD} \tag{M9}$$

Combining Eqs. (M8) and (M9), the ratio of vegetated to bare bed critical Shields parameter ($\theta_{c,n}/\theta_{c,0}$) or critical shear stress ($\tau_{c,n}/\tau_{c,0}$) is[29]:

$$\frac{\theta_{c,n}}{\theta_{c,0}} = \frac{\tau_{c,n}}{\tau_{c,0}} = \frac{U_{c,n}^2}{U_{c,0}^2} \tag{M10}$$

Using Eq. (M10), the critical shear stress for a marsh with stem density $n$, $\tau_{c,n}$, was estimated from the critical shear stress for a bare bed with the same sediment and used in Eq. (M7).

**Model set-up**
This study simulated the flux of water and sediment from a channel onto a marsh platform with different densities of emergent vegetation based on typical conditions of marshes collected from the literatures[18–21]. The flow entering the marsh platform was driven by the water surface slope ($S$) between the channel and marsh platform. Flow direction was assumed to be perpendicular to the marsh edge. For the base cases, $S = 0.0005$ and water depth $H = 0.3$ m. The grain size of marshes generally ranges from $D = 10$ to $70$ µm[18–21]. For simplicity, we considered uniform silt with $D = 50$ µm ($\rho_s = 2650$ kg/m³). The settling velocity, $w_s$, was estimated using the Ferguson and Church formulation[60]. For $D = 50$ µm, $w_s = 2.1$ mm/s, which falls in the range for mud from the previous studies[61]. The bed-drag coefficient, $C_f$, was set to a spatially and temporally constant value of $0.005$[20].

Common vegetation species in the marshes have stem diameters $d$ ranging from 0.5 cm to 1.5 cm[18–21]. A diameter of 1 cm was chosen in the

base cases. The vegetation density, $n$, varied from 0 to 500 stems/m$^2$ (16 different values), which spans values observed in the field. The vegetation drag coefficient $C_d$ was taken as 1.0[62]. For simplicity, the vegetation on the marsh was assumed to be emergent, such that the vegetation height in the model was set to the depth $H$. To capture attributes of different estuaries and marshes and the seasonal effects of vegetation growth and water flux, additional slopes ($S = 0.001$ and 0.00025) and stem diameters ($d = 0.5$ and 1.5 cm) were considered.

In the RDM, particles were released continuously and uniformly over the depth at the marsh edge ($x = 0$, Fig. 1), and the model was run for 1000 s, which ensured that the sedimentation rate (i.e., net deposition rate $q_d$) reached an equilibrium state (Supplementary, S1). Simulations with different particle numbers, $N$, and time-step, $\Delta t$, ranging from 10,000 to 100,000 and 0.01 to 0.1 s, respectively, confirmed that the solution was insensitive to particle number and time-step within this range (varied by less than 6.5%). A particle number of 50,000 and a time-step of 0.05 s were adopted to satisfy stability criteria. Each case was repeated for ten realizations, and the ensemble average was used in the reported results. As an example of statistical variation, the ensemble standard deviation is shown with gray shading in Fig. 3b.

### Field data validation

Field data from Beltrán-Burgos[19] was compared to model simulations. The study measured velocity, sediment, and vegetation parameters at different stages during the growing season between April and August 2019 within the Cubits Gap sub-delta in the MRD (Fig. 1b, c). Details of the field measurements can be found in Chapter 3 in ref. 19. The vegetation at the mudflat included *Potamogeton nodosus*, *Stuckenia pectinate*, and *Potamogeton crispus*. For simplicity, we based the model vegetation on the dominant species *Potamogeton nodosus*, which has oval leaves (-15 cm long and -2 cm wide). Vegetation coverage, $V_c$, defined as the fraction of horizontal surface area obstructed by plants, was estimated on April 24, June 3, and July 1, using random quadrat (0.25 m$^2$) surveys and recreated species, *Potamogeton malainus*, ref. 63 correlated $V_c$ measured with underwater digital imaging to the one-side leaf area index (LAI, surface area per bed area). The empirical relation (See Fig. 8 in ref. 63) was applied in the present study. LAI is a reasonable estimate for plant frontal area per bed area, such that $ndH = $ LAI in Eq. (M4). The measured velocity and LAI were used in Eq. (M4) to estimate water surface slope, $S$. In *Potamogeton* species, the stems are small compared to the leaves, such that the leaf width was assumed to determine the turbulence length-scales, i.e., $l_t = l_w$. Turbulence length-scale defined by leaf width was also observed for *Typha* in Xu and Nepf[27]. Since the leaves contributed the majority of plant frontal area[63], we defined vegetation density based on leaf area, i.e., $n_l$ (leaves/m$^2$) = LAI/$A_l$, with $A_l$ the area of one leaf.

The sedimentation rate, $q_d$, was defined by the total number of particles deposited, $M_{nd,n}$, over domain area, $A$, and simulation time, $T$, i.e.,

$$q_d = \frac{M_{nd,n}}{A \cdot T}. \tag{M11}$$

Because the lateral dimension, and thus $A$, is not explicitly defined within the model, a direct comparison between RDM and measured $q_d$ would only be possible through calibration. To eliminate the need for calibration, the measured and simulated sedimentation rates, $q_d$, were normalized by their maximum value in July, i.e., $\widetilde{q_d} = q_d/q_{dJuly}$. The normalized sedimentation rates are shown in Fig. 5.

### Data availability
The field data in this study are extracted from the study[19]. The data generated in this study have been deposited in the Zenodo database (https://doi.org/10.5281/zenodo.6754621).

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

## Acknowledgements

This study received support from the National Natural Science Foundation of China (51879138), US NSF Grant EAR 1854564, and the National Academies of Sciences, Engineering, and Medicine under Grant Agreement number 2000008941. The authors thank Elizabeth Follett for her assistance with the numerical model. The authors also thank Danxun Li, Liecai Cao, and Yanchong Duan for their discussions.

## Author contributions

H.M.N. and Y.X. designed the study. Y.X. and H.M.N. built the model. C.R.E. and M.B.-B. contributed field insights and field data. Y.X. drafted the paper. All authors contributed to the interpretation of the results and editing of the text.

## Competing interests

The authors declare no competing interests.
