## [Peer Review File · Nature Communications]

Competing effects of vegetation density on sedimentation in deltaic marshesReviewers' Comments:

Reviewer #1:

Remarks to the Author:

Synthesis of the study.

The study by Yuan Xu et al. aims to understand sediment transport in a riverine marsh through a random displacement model application. The authors applied the model to analyze different eco-geomorphic settings which might affect sediment deposition into the marsh canopy. Authors explored with numerical experiments effects of a) vegetation density on flow conditions; b) vegetation on sedimentation; c) vegetation on sedimentation spatial distribution. Then, they described models applied in the proposed study and its setting. The authors submitted a supplemental material document with a clear description of the numerical model.

Comments.

The study is well designed and results are promising however, the research is missing the validation and calibration of the model. My main suggestion is to re-organize the manuscript and add the important part of validation and calibration which can be done with previous studies and measurements available in the literature.

All the sections are not well-connected. If Authors would write a manuscript focused entirely on numerical modeling, some parts of the explanations in the supplemental material might be included in the main text. The title and abstract are promising but reading the manuscript the study loses intensity. I suggest resubmitting this manuscript once the text has been re-organized, the model validated/calibrated and results are connected.

One of my major concerns is referred to the initial and boundary conditions of the model. Did the Authors check the influence of flow direction through the marsh, especially at the marsh edge? Is the flow perpendicular to the marsh? If yes, please explain it. If not, please add some text in the discussion on how to include it and what did you underestimate.

Figure 1 shows a real marsh example for model scenarios. However, it's not clear why the vegetation changed so much during those two periods. I can envision different reasons for these changes: increased water level, marsh die-off, effects of seasonality changes. I think the Authors should include a little description and discussion about it.

In conclusion, the supplemental material looks great with a lot of model insight. I enjoyed reading it and I would suggest adding some description in the main text. It will help the reader to follow the manuscript.

Reviewer #2:

Remarks to the Author:

The article investigates the impact of vegetation density on the sediment redistribution on a marsh platform. The article uses a random displacement model and the input parameters for this are derived from a dataset from the Mississippi River Delta.

The article is of interest for experts in the subject and the overall approach, while simplified, is rigorous. In my opinion, the article falls short in terms of the significance of the main outcomes.

The topic has been widely investigated and the outcomes here mostly confirm previous results from modelling and field investigations in terms of sedimentation patterns and magnitude as well as vegetation dynamics. This is good and lends credibility to the authors' results. However, it also leaves a reader wondering about what is the real novelty and significance which is relevant for publication in this journal.

As a second minor concern, I find the description of the model confusing. For instance, the reference to figure 1 for model setup can be misleading. This could have probably been clearer in the case of an

expanded methodology section in more traditional Journals. The final portion of the abstract stating that the results agree with field observation could also appear misleading if not explained better as the portion of the manuscript dealing with the field data comparison is extremely slim and mostly dedicated to the validation of model inputs rather than a comparison of the outputs.

Reviewer #3:

Remarks to the Author:

This manuscript uses field-informed numerical modeling to evaluate how marsh architecture and water-surface slope impact sedimentation, applying the most rigorous and current physical representations of how vegetation impacts flow and turbulence. As with a paper previously published in *Nature Geoscience* (Nardin & Edmonds, 2014, "Optimum vegetation height and density for inorganic sedimentation in deltaic marshes," this manuscript finds that intermediate vegetation densities lead to the greatest sedimentation. When I started reading this manuscript, I felt conflicted about my recommendation given the previous finding of Nardin and Edmonds, but I do think this paper makes some important advances over that one. First, the modeling of vegetation and its effects on sediment is much more rigorous in the current manuscript compared to Nardin & Edmonds, where vegetation is represented through a Manning's n (which is problematic for vegetation), and turbulence intensity and resuspension are not accounted for. Second, the evaluation of the sensitivity of water surface slope in the present paper is a useful advance that result in some interesting (even surprising) differences between low-flow and high-flow regimes. Third, this paper revisits an assumption commonly invoked in marsh geomorphology—that deposition declines exponentially with distance from the channel—and finds that it only applies in certain special cases.

Results of this paper will have impacts on planning marsh restoration in Louisiana and may also yield improvements in how sedimentation is modeled in investigations of marsh persistence in the face of sea-level rise. It will be of interest to hydrodynamicists, fluvial geomorphologists, coastal engineers, and marsh ecologists.

In general, the manuscript is very well written and organized, with a few minor exceptions. First, the justification for this study—and particularly how it moves beyond Nardin & Edmonds—could be improved. Lines 40-41 state that the role of vegetation density has been previously examined through modeling, but there have been few field studies, which leads reader to think that this study will be a field study. However, that is not the case. While field sedimentation results are referred to (lines 209-2016), none are actually presented (see comment below about how having access to some of the actual data would be desirable), and the real advance of this work lies in the modeling.

Secondly, I would like to see justification of why an RDM is used for sediment transport rather than a mass-balance model, as is common in other marsh modeling studies. I was unfamiliar with RDMs prior to reading the manuscript and am genuinely curious about why this approach was chosen.

Third, perhaps this question is naïve given my unfamiliarity with RDM models, but it seems that the stochasticity inherent in how sediment transport is modeled would generate an envelope of predictions (as in Figure S1), yet no uncertainty is depicted in the main figures. If these findings are to be used by restoration practitioners, including estimates of uncertainty—particularly with respect to the optimum density—seems important.

LINE-BY-LINE/SPECIFIC COMMENTS

Title: The title doesn't make sense to me. It seems like there is a word missing. Should there be an "on" before "sedimentation?" If this is the case, I would also recommend that the word "control" be eliminated, as it doesn't add anything and makes the title more confusing.

Line 45: It would help if the RDM were briefly defined/described where it is first introduced, to make this paper more accessible to readers.

~Line 48: It is not clear from the outset what the scale of the domain is. Fig. 1 implies a fairly small scale surrounding the marsh platform/channel interface, but no dimensions are given.

Figure 2: This figure was really confusing to me, in part because the same colors are used for cases that share no similarity in parameters. I suggest making interpretation easier on the readers by using a particular "point" convention to discriminate between slopes and color convention to discriminate between diameters. The third sentence of the caption is also very confusing, as no overlap is actually depicted, and the parentheses refer to just one gray line for "cases with the same stem diameter but different slopes," which actually refers to the three cases in the inset. If that sentence is corrected, the following one (lines 86-87) would become redundant and could be removed.

Line 95: I don't believe phi has been defined yet.

Line 96: Add "in" after "variation."

Line 122: Comma needed after "resuspension."

Line 149: Suggest revising end of sentence to "but has a smaller magnitude."

Figure 3: In part e, I do not see the blue or red line with the triangle points. Are they behind the gray line?

Line 191: captures \diamond captured

Lines 209-216: Would be nice to see some data associated with the field studies in the main paper, perhaps in a supplemental section.

Line 225-226: I would be interested in how the authors justify their choice of an RDM over a more standard mass-balance sediment transport model.

Response to Reviewers

We thank the reviewers for their helpful suggestions. Their constructive comments considerably strengthened the paper. In the manuscript modifications are highlighted by blue font. In the response below, black font, blue font, and red font denote reviewers' comments, response to reviewers' comments, and modified content in the revised manuscript.

Response to Reviewer #1

Synthesis of the study.

The study by Yuan Xu et al. aims to understand sediment transport in a riverine marsh through a random displacement model application. The authors applied the model to analyze different eco-geomorphic settings which might affect sediment deposition into the marsh canopy. Authors explored with numerical experiments effects of a) vegetation density on flow conditions; b) vegetation on sedimentation; c) vegetation on sedimentation spatial distribution. Then, they described models applied in the proposed study and its setting. The authors submitted a supplemental material document with a clear description of the numerical model.

Comments.

The study is well designed and results are promising however, the research is missing the validation and calibration of the model. My main suggestion is to re-organize the manuscript and add the important part of validation and calibration which can be done with previous studies and measurements available in the literature.

Response: We agree that the validation of the model is important and should be clearly described. Some aspects of the modeling were validated in a previous study (Xu & Nepf, 2021), which is now cited in line 57. In addition, a detailed comparison to field measurements has been added, described in a new Results section, "Validation with field measurements" and a new section in the Methods, "Field data validation". For your convenience, the content is repeated here:

Validation with field measurements

The simulation of net deposition was validated by comparison to a field study reported in Burgos¹⁹, which measured velocity, sediment, and vegetation parameters over a growing season at Cubits Gap, a sub-delta in the MRD (Fig. 1b and 1c). At this site, the maximum sedimentation occurred during a period of intermediate vegetation density (July). The model inputs, based on the field measurements, are summarized in Table 1

and described in the Methods. The measured (star) and modeled (circle) sedimentation rates, \tilde{q}_d , had good agreement (Fig. 5), indicating that the RDM captured the competing effects of supply restriction and retention efficiency to predict the observed non-linear dependence between vegetation density and net deposition.

Table 1 Field data in Cubits Gap sub-delta

	H (m)	U (cm/s)	V_c (%)	LAI (m ² /m ²)	n_l (leaves/m ²)	S	q_d (g/m ² /day)
April 24	0.4	13	5	0.05	16	0.00013	14
June 3	0.6	3	66	1.02	340	0.00008	190
July 1	0.6	6	20	0.21	70	0.00007	324

Velocity, U , from drone images of tracer motion, and vegetation coverage, V_c , measured on dates shown in the table. Leaf area index, LAI (one-sided leaf area per bed area) estimated from V_c using method described in ref.³⁴ (See Methods). The leaves per bed area, $n_l = LAI/A$, with A the one-sided area of one leaf. The water surface slope, S , was not measured in the field study, but was estimated from Eq. (M4) and the measured velocity, using $ndH = LAI$. Sedimentation rate, q_d , for each data was estimated from net deposition recorded on tiles just before and just after the vegetation and velocity measurement dates (see Fig. 41b mudflat site in Burgos¹⁹). Grain size in the field was between 12 and 53 μm , and in the model was set to 50 μm .

Fig. 5 | Comparison between measured (stars) and simulated (circles) sedimentation rate. Grey dashed curve added to emphasize non-linear relationship between net deposition rate and vegetation density.

Field data validation

Field data from Burgos¹⁹ was compared to model simulations. The study measured velocity, sediment, and

vegetation parameters at different stages during the growing season between April and August 2019 within the Cubits Gap sub-delta in the MRD (Fig. 1b and 1c). Details of the field measurements can be found in Chapter 3 in ref.¹⁹. The vegetation at the mudflat site included *Potamogeton nodosus*, *Stuckenia pectinate*, and *Potamogeton crispus*. For simplicity, we based the model vegetation on the dominant species *Potamogeton nodosus*, which has oval leaves (~15 cm long and ~2 cm wide). On April 24, June 3, and July 1, random quadrat (0.25 m²) surveys and rectified drone photographs were used in ref.¹⁹ to estimate vegetation coverage, V_c , defined as the fraction of horizontal surface area obstructed by plants. Considering a related species, *Potamogeton malainus*, Zhao et al.³⁴ correlated V_c measured with underwater digital imaging to the one-side leaf area index (LAI , surface area per bed area). The empirical relation (See Fig. 8 in Zhao et al.³⁴) was applied in the present study. LAI is a reasonable estimate for plant frontal area per bed area, such that $ndH = LAI$ in Eq. (M4). The measured velocity and LAI were used in Eq. (M4) to estimate water surface slope, S . In *Potamogeton* species, the stems are small compared to the leaves, such that the leaf width was assumed to determine the turbulence length-scales, i.e., $l_t = l_w$. Turbulence length-scale defined by leaf width was also observed for *Typha* in Xu and Nepf²⁷. Since the leaves contributed the majority of plant frontal area³⁴, we defined vegetation density based on leaf, i.e., n_l (leaves/m²) = LAI/A_l , with A_l the area of one leaf.

The sedimentation rate, q_d , was defined by the total number of particles deposited, $M_{nd,n}$, over domain area, A , and simulation time, T , i.e.,

$$q_d = \frac{M_{nd,n}}{A \cdot T} . \quad (M11)$$

Because the lateral dimension, and thus A , is not explicitly defined within the model, a direct comparison between RDM and measured q_d is possible through calibration. Alternatively, the measured and simulated sedimentation rates, q_d , were normalized by their maximum value in July, i.e., $\tilde{q}_d = q_d/q_{d,July}$. The normalized deposition rates are shown in Fig. 5.

All the sections are not well-connected. If Authors would write a manuscript focused entirely on numerical modeling, some parts of the explanations in the supplemental material might be included in the main text. The title and abstract are promising but reading the manuscript the study loses intensity. I suggest resubmitting this manuscript once the text has been re-organized, the model validated/calibrated and results are connected.

Response: Thank you. We intended the paper to go beyond a description of the modeling and to discuss the insight provided by the model regarding the effects of vegetation on sedimentation and its spatial distribution in the marsh. However, we now see that we pushed too much of the modeling description to the supporting

information. We agree that more of the modeling framework should be included in the main text. We have moved supplementary S1 and S2 (RDM description in previous manuscript) to Methods and added more description of the model in Methods (Please see “Modeling method” in the revised version, Line 256 to 334).

More importantly, we have added a new validation of the model using field data. Please see “Validation with field measurements” (Line 200) in Results and “Field data validation” (Line 361) in Methods.

One of my major concerns is referred to the initial and boundary conditions of the model. Did the Authors check the influence of flow direction through the marsh, especially at the marsh edge? Is the flow perpendicular to the marsh? If yes, please explain it. If not, please add some text in the discussion on how to include it and what did you underestimate.

Response: The model assumes the flow enters the marsh in a direction perpendicular to the channel-marsh edge. The flow is steady and driven by a water surface slope between the channel and marsh platform. We have added this at Line 60. We also marked flow direction in Fig. 1a. Finally, we added the following discussion (Line 271):

“It is worth noting that the 2-D model is valid along any vector parallel to the flow direction entering the marsh, as long as the transport onto the marsh platform is dominated by advection. Situations in which the exchange with the marsh is predominantly by lateral diffusion would not be represented by this model. For simplicity, in this study flow direction is assumed to be perpendicular to the marsh edge (Fig. 1), and lateral diffusion is not considered.”

Figure 1 shows a real marsh example for model scenarios. However, it’s not clear why the vegetation changed so much during those two periods. I can envision different reasons for these changes: increased water level, marsh die-off, effects of seasonality changes. I think the Authors should include a little description and discussion about it.

Response: Thank you for this suggestion. Changes in vegetation coverage were due to the natural variation over the growing season. Specifically, vegetation bloomed in June and died off in July. We have added this explanation to the caption of Fig.1 (Line 71).

“Change in vegetation coverage was due to natural variation over the growing season.”

In conclusion, the supplemental material looks great with a lot of model insight. I enjoyed reading it and I would suggest adding some description in the main text. It will help the reader to follow the manuscript.

Response: We are glad the reviewer enjoyed reading our study. As suggested, we have moved more description of the model into the main text to help the reader to follow the manuscript. See lines 256 to 279 and 295 to 334.

Response to Reviewer #2

The article investigates the impact of vegetation density on the sediment redistribution on a marsh platform. The article uses a random displacement model and the input parameters for this are derived from a dataset from the Mississippi River Delta. The article is of interest for experts in the subject and the overall approach, while simplified, is rigorous. In my opinion, the article falls short in terms of the significance of the main outcomes.

Response: We thank the reviewer for the helpful suggestions for further improvements. In the revised version we clarified the novelty and significance of the work (see Line 40 to 58). More details are provided in the comments below.

The topic has been widely investigated and the outcomes here mostly confirm previous results from modelling and field investigations in terms of sedimentation patterns and magnitude as well as vegetation dynamics. This is good and lends credibility to the authors' results. However, it also leaves a reader wondering about what is the real novelty and significance which is relevant for publication in this journal.

Response: We agree that the novelty of this study was not stated clearly in the previous manuscript. This study offers several novel details, which were also highlighted by Reviewer #3, i.e.,

- The modelling of vegetation and its effects on sediment movement is described with a more rigorous physical bases than previous studies. Most previous models describe vegetation effects only through a modified Manning's roughness, which does not account for the additional turbulence generated by vegetation and its impact of sediment movement. The model in the present study explicitly includes the impact of vegetation-generated turbulence on both the turbulent diffusivity and also on the critical bed-shear stress.
- In contrast to previous studies, which typically consider a limited number of specific, field-scale scenarios, this study intentionally considered a reduced-order (2-D) representation of the marsh-channel boundary in order to explore a wide range of conditions. This enabled the identification of regime shifts in sedimentation pattern and the parameters that define them (e.g., Fig. 3 and 4 in the manuscript). In particular, this framework gave new insight into an assumption commonly invoked in marsh geomorphology—that deposition declines exponentially with distance from the channel—and to show that this assumption only applies in certain conditions.
- The exploration of different water surface slope reveals the differences between low-flow and high-flow regimes.

We have edited the paper to explicitly clarify the novelty and significance of our study, e.g.,

Starting at Line 48: This study considered the impact of vegetation density on the sediment supply to and retention on a marsh platform using a random displacement model (RDM), a Lagrangian method that tracks the transport of individual sediment particles subject to advection, diffusion, deposition, and entrainment. The RDM method was chosen for its computational efficiency. Unlike most numerical models, which represent vegetation through a Manning's roughness^{13,20,21}, our model considered the effects of vegetation size and area density on velocity, turbulence, and diffusivity, each of which influences sediment deposition and resuspension. Rather than considering a specific site, this study intentionally considered a reduced-order (2-D) representation of a marsh platform in order to explore a wide parameter space. The RDM method enabled us to separately quantify the effects of supply and retention and their impact on sedimentation and its spatial distribution. A previous study validated RDM modeling of sediment transport and suspended sediment concentration using laboratory measurements²². The present study offers further validation through a comparison to field measurements.

Starting at Line 248: Using RDM simulation to consider the interaction between flow, vegetation, and sediment, this study provides important insight into how vegetation density (related to species and growth phase) influences the potential to trap sediment and build marsh, which are key processes needed to save drowning coasts, such as Louisiana, USA. In particular, the study can provide guidance to optimize the retention of sediment by wetlands targeted by sediment diversions, both at the scale of local levee cuts and by the billion-dollar Mid-Barataria and Mid-Breton structures.

As a second minor concern, I find the description of the model confusing. For instance, the reference to figure 1 for model setup can be misleading. This could have probably been clearer in the case of an expanded methodology section in more traditional Journals. The final portion of the abstract stating that the results agree with field observation could also appear misleading if not explained better as the portion of the manuscript dealing with the field data comparison is extremely slim and mostly dedicated to the validation of model inputs rather than a comparison of the outputs.

Response: First, we agree that the original Fig. 1 was misleading with regard to model set up, as it implied a connection to a specific site. The model was not constructed for a specific site, but constructed to explore a wide range of conditions. To clarify this, we added a cartoon schematic of the modelled scenario (Fig. 1a, below). In addition, we moved some details of the model from the supporting information to the methods in the main text. Please see "Modeling method" in the revised version, Line 256 to 334.

Fig. 1a Schematic of modelled scenario

Second, we agree with that the field data comparison was extremely slim in the previous manuscript. To correct this, we have added a more detailed description, including a figure. Please see Figure 5 in the new “Validation with field measurements” (Line 200) and “Field data validation” (Line 361) in Methods.

Response to Reviewer #3

This manuscript uses field-informed numerical modeling to evaluate how marsh architecture and water-surface slope impact sedimentation, applying the most rigorous and current physical representations of how vegetation impacts flow and turbulence. As with a paper previously published in *Nature Geoscience* (Nardin & Edmonds, 2014, “Optimum vegetation height and density for inorganic sedimentation in deltaic marshes,” this manuscript finds that intermediate vegetation densities lead to the greatest sedimentation. When I started reading this manuscript, I felt conflicted about my recommendation given the previous finding of Nardin and Edmonds, but I do think this paper makes some important advances over that one. First, the modeling of vegetation and its effects on sediment is much more rigorous in the current manuscript compared to Nardin & Edmonds, where vegetation is represented through a Manning’s n (which is problematic for vegetation), and turbulence intensity and resuspension are not accounted for. Second, the evaluation of the sensitivity of water surface slope in the present paper is a useful advance that result in some interesting (even surprising) differences between low-flow and high-flow regimes. Third, this paper revisits an assumption commonly invoked in marsh geomorphology—that deposition declines exponentially with distance from the channel—and finds that it only applies in certain special cases.

Results of this paper will have impacts on planning marsh restoration in Louisiana and may also yield improvements in how sedimentation is modeled in investigations of marsh persistence in the face of sea-level rise. It will be of interest to hydrodynamicists, fluvial geomorphologists, coastal engineers, and marsh ecologists.

Response: We thank the reviewer for these positive comments.

In general, the manuscript is very well written and organized, with a few minor exceptions. First, the justification for this study—and particularly how it moves beyond Nardin & Edmonds—could be improved. Lines 40-41 state that the role of vegetation density has been previously examined through modeling, but there have been few field studies, which leads reader to think that this study will be a field study. However, that is not the case. While field sedimentation results are referred to (lines 209-216), none are actually presented (see comment below about how having access to some of the actual data would be desirable), and the real advance of this work lies in the modeling.

Response: Thank you. First, we agree that the introduction should clearly state how the present study moves beyond Nardin and Edmonds (2014). To improve this, we have added new text to the Introduction (at Line 40).

Nardin and Edmonds¹³ first described these competing influences of vegetation, enhanced retention and reduced supply, showing that both can impact the net accumulation of sediment on a marsh platform. At present there are few studies that consider both processes or quantify their competing effects on sedimentation and its spatial distributions. In addition, most model studies, e.g., Nardin and Edmonds¹³, Olliver et al.²¹, simplify the vegetation effects through a Manning's roughness, which does not account for the additional turbulence generated by vegetation, which can also impact sediment transport²²⁻²⁴.

Understanding the influence of vegetation on the competing mechanisms of sediment supply and retention are critical for planning successful restoration strategies that will recover lost land. This study considered the impact of vegetation density on the sediment supply to and retention on a marsh platform using a random displacement model (RDM), a Lagrangian method that tracks the transport of individual sediment particles subject to advection, diffusion, deposition, and entrainment. The RDM method was chosen for its computational efficiency. Unlike most numerical models, which represent vegetation through a Manning's roughness^{13,20,21}, our model considered the effects of vegetation size and area density on velocity, turbulence, and diffusivity, each of which influences sediment deposition and resuspension. Rather than considering a specific site, this study intentionally considered a reduced-order (2-D) representation of a marsh platform in order to explore a wide parameter space. The RDM method enabled us to separately quantify the effects of supply and retention and their impact on sedimentation and its spatial distribution. A previous study validated RDM modeling of sediment transport and suspended sediment concentration using laboratory measurements²². The present study offers further validation through a comparison to field measurements.

Second, we see how the phrasing of the noted sentence implies that the present study is a field study. We have removed the adjective "field" to avoid this. Original sentence: "At present there are scant field studies that consider both processes or quantify their competing effects on sedimentation" New sentence: "At present there are few studies that consider both processes or quantify their competing effects on sedimentation and its spatial distributions" (Line 42).

Third, we have expanded the comparison to field data collected by Burgos (2021). We have added a more detailed description, including a figure. Please see Fig. 5 in the new section in Results, "Validation with field measurements" (Line 200) and the new section in methods, "Field data validation" (Line 361).

Reference:

Nardin, W., & Edmonds, D. A. (2014). Optimum vegetation height and density for inorganic sedimentation in deltaic marshes. *Nature Geoscience*, 7(10), 722–726. <https://doi.org/10.1038/ngeo2233>

Burgos, M. B. (2021). Effects of vegetation seasonality on sediment dynamics in a freshwater marsh of the Mississippi River Delta (PhD thesis). LA, USA: Tulane University.

Secondly, I would like to see justification of why an RDM is used for sediment transport rather than a mass-balance model, as is common in other marsh modeling studies. I was unfamiliar with RDMs prior to reading the manuscript and am genuinely curious about why this approach was chosen.

Response: Eulerian (mass-balance) transport models suffer from numerical dispersion, which is overcome by making grid and time steps smaller, which results in long computational run times. RDM (Lagrangian transport) models do not suffer from numerical dispersion and thus can be more computationally efficient. RDM models are common in aquifer and terrestrial canopy studies, and the Nepf lab began developing the present model to study spore transport in terrestrial crop canopies (e.g., Follett et al., 2016). The model was later modified to simulate suspended sediment transport in *Typha latifolia* (Xu & Nepf, 2021). Because the RDM model is computationally efficient, it enables the evaluation of a large number of cases to explore a wide range of conditions, which was the focus of the present study.

Reference:

Follett, E., Chamecki, M., & Nepf, H. (2016). Evaluation of a random displacement model for predicting particle escape from canopies using a simple eddy diffusivity model. *Agricultural and Forest Meteorology*, 224, 40–48. <https://doi.org/10.1016/j.agrformet.2016.04.004>

Xu, Y., & Nepf, H. (2021). Suspended Sediment Concentration Profile in a *Typha Latifolia* Canopy. *Water Resources Research*, 57(9), e2021WR029902. <https://doi.org/10.1029/2021WR029902>

Third, perhaps this question is naïve given my unfamiliarity with RDM, but it seems that the stochasticity inherent in how sediment transport is modeled would generate an envelope of predictions (as in Figure S1), yet no uncertainty is depicted in the main figures. If these findings are to be used by restoration practitioners, including estimates of uncertainty—particularly with respect to the optimum density—seems important.

Response: Yes. We thank the reviewer for raising this issue, which should have been explained more clearly. To estimate the uncertainty of RDM results, each simulation was repeated for ten realizations. The final result is the ensemble average of ten realizations. The standard deviation among the ten realizations is a metric of uncertainty. As an example, we added the ensemble standard deviation to Fig. 3b (shown below),

and this is now described in the caption of Fig. 3, i.e., “The gray shading around ND curve denotes the ensemble standard deviation based on ten realizations (Methods), which ranged from 1.1% to 6.0% of the ensemble average”. We also added a note at Line 356 in the Methods.

Fig. 3b Modelled sedimentation (net deposition) with vegetation area density n

LINE-BY-LINE/SPECIFIC COMMENTS

Title: The title doesn't make sense to me. It seems like there is a word missing. Should there be an “on” before “sedimentation?” If this is the case, I would also recommend that the word “control” be eliminated, as it doesn't add anything and makes the title more confusing.

Response: We revised the title to

“Competing effects of vegetation density on sedimentation in deltaic marshes”.

Line 45: It would help if the RDM were briefly defined/described where it is first introduced, to make this paper more accessible to readers.

Response: Thank you for this suggestion. We have added a brief description of RDM when it is first introduced. At line 48 “This study considered the impact of vegetation density on the sediment supply to and retention on a marsh platform using a random displacement model (RDM), a Lagrangian method that tracks the transport of individual sediment particles subject to advection, diffusion, deposition, and entrainment. The RDM method was chosen for its computational efficiency.”.

~Line 48: It is not clear from the outset what the scale of the domain is. Fig. 1 implies a fairly small scale surrounding the marsh platform/channel interface, but no dimensions are given.

Response: We added a schematic of the modelled scenario as Fig. 1a in the revised manuscript, and it includes the scale of the domain.

Fig. 1a Schematic of modelled scenario

Figure 2: This figure was really confusing to me, in part because the same colors are used for cases that share no similarity in parameters. I suggest making interpretation easier on the readers by using a particular “point” convention to discriminate between slopes and color convention to discriminate between diameters. The third sentence of the caption is also very confusing, as no overlap is actually depicted, and the parentheses refer to just one gray line for “cases with the same stem diameter but different slopes,” which actually refers to the three cases in the inset. If that sentence is corrected, the following one (lines 86-87) would become redundant and could be removed.

Response: Thank you for these valuable suggestions. We now use color to discriminate between diameters and point shape to discriminate between slopes. For your convenience, we attach the revised figure here. We have also revised the confusing sentence (line 101) “The normalized velocity is not a function of slope (Eq. (M4)), so that all cases with same stem diameter but different slope collapse to a single curve, shown by gray curve. The inset graph shows the dimensional velocity U_n for different slopes with $d = 1.0 \text{ cm}$ ”.

Fig. 2 Flow parameters with vegetation area density n

Line 95: I don't believe phi has been defined yet.

Response: Thanks. We have added the definition of ϕ . At line 111 “However, increasing d also increases solid volume fraction, ϕ , which tends to enhance TKE¹⁵.”

Line 96: Add “in” after “variation.”

Response: Thanks. We have added “in” after “variation”. Line 113

Line 122: Comma needed after “resuspension.”

Response: Thanks. We have added comma “,” after “resuspension”. Line 122.

Line 149: Suggest revising end of sentence to “but has a smaller magnitude.”

Response: Thanks. The sentence, now at line 167, has been revised to “Consequently, the peak progressively shifts to a larger n , but has a smaller magnitude”.

Figure 3: In part e, I do not see the blue or red line with the triangle points. Are they behind the gray line?

Response: Yes. Because sediment supply ratio is not a function of slope (See Eqs. (1) and (M4)), the three lines with different slopes collapse to a single curve (gray line with triangle). To clarify this, we have added the explanation in the caption of Figure 3, i.e., “**In e note that SSR is not a function of slope (Eqs. (1) and (M4)), so that SSR curves with different slope overlap, shown by gray line with triangle.**”

Line 191: captures \diamond captured

Response: Thanks. We have revised to “captured” – Line 231.

Lines 209-216: Would be nice to see some data associated with the field studies in the main paper, perhaps in a supplemental section.

Response: Thank you for this suggestion. We have added a more detailed description, including a figure. Please see Fig. 5 in the new section in Results, “Validation with field measurements” (Line 200) and the new section in methods, “Field data validation” (Line 361).

Line 225-226: I would be interested in how the authors justify their choice of an RDM over a more standard mass-balance sediment transport model.

Response: RDM (Lagrangian transport) models do not suffer from numerical dispersion and are more computationally efficient than the Eulerian mass-balance model. For this reason, RDM models have become common in aquifer and terrestrial canopy studies. The Nepf lab began developing the present model to study spore transport in terrestrial crop canopies (e.g., Follett et al. 2016). The model was later modified to simulate suspended sediment transport in *Typha latifolia* (Xu & Nepf, 2021). Because the model is computationally efficient, it can be used to quickly compute a large number of cases to explore a wide range of conditions, which was the focus of the present study.

Reviewers' Comments:

Reviewer #1:

Remarks to the Author:

The manuscript has been improved with this first round of review. I appreciated the better description of field data validation and other description along with the discussion.

There are still some typos in the text. I would suggest adding commas in numbers such as 1,000 instead of 1000 to help the reader understand numbers.

I suggest publication of the manuscript after minor review relative to not significant typos.

Reviewer #2:

Remarks to the Author:

The authors have addressed my main concerns and I think the clarity is improved including an improvement in the delivery of the main message.

The main outcome is that there are competing trends when considering sediment deposition (supply and retention) which produce a nonlinear relationship between net deposition and vegetation density. I think the work could have some potential for influencing restoration projects though in order for that to really happen a wider analysis in terms of general geomorphic settings would be needed. The latter would be outside the scope of the work so I would not ask the authors to address this unless they think some mention of this could serve to enrich the discussion section. The main analyses support the conclusions. The model validation remains a little bit slim and a minor suggestion would be to try finding more data points for validation or trying to replace the grey dashed curve in figure 5 with an ensemble of simulation points. This would improve the quantitative nature of the results. However, this is a suggestion, and the main message of the article could be delivered without implementing this.

Reviewer #3:

Remarks to the Author:

First, the authors did a fantastic job addressing the reviewer comments, and the manuscript is much improved. I remain enthusiastic about its contribution, and my remaining comments are relatively minor. The most substantive of my comments is that some of what I see as some of the most important advances of the paper, regarding the finding that the exponential deposition profile applies only to special cases and, perhaps, the finding about the importance of water-surface slope, are relatively buried. I would recommend adding a sentence about the exponential deposition profile to the abstract, at a minimum, so that it better reflects the novel contributions of the paper. Likewise, the Discussion focuses primarily on exceptions to the model (i.e., shortcomings, caveats, and discrepancies between the model and observations), rather than highlighting the key contributions of the work and their implications. It seems that at least one opening paragraph that discusses the positive and novel contributions of the model (such as implications of the finding that the exponential deposition model applies only to select cases, or implications of the finding that at low water surface slopes, vegetation does not yield more deposition, and what this means in places where flow has been impounded or otherwise reduced) would improve the organization of this paper. A final organizational comment is that it seems strange to present equations 1-3 in the Results, when this text is really methods.

Additional editorial comments:

Lines 14-15: Consider rewriting as "Marsh vegetation, a definitive component of delta ecosystems,..."

Line 47: are  is

Line 153: Add "of" after "most."

Line 367-369: This seems to be a run-on sentence, and additionally I believe a verb is missing.

Line 373: leave  leaf

Line 376: leave  leaf area

Response to Reviewers

We thank the reviewers for their valuable suggestions. In the manuscript, modifications are highlighted by blue font. In the response below, black font, blue font, and red font denote reviewers' comments, response to reviewers' comments, and modified content in the revised manuscript.

Response to Reviewer #1

The manuscript has been improved with this first round of review. I appreciated the better description of field data validation and other description along with the discussion.

Response: We thank the reviewer for their appreciation of our work and helpful suggestions.

There are still some typos in the text. I would suggest adding commas in numbers such as 1,000 instead of 1000 to help the reader understand numbers.

Response: Thank you. We have reviewed the manuscript carefully and corrected the typos. Additionally, we have also added the commas in numbers.

I suggest publication of the manuscript after minor review relative to not significant typos.

Response: We have reviewed the manuscript for typos.

Response to Reviewer #2

The authors have addressed my main concerns and I think the clarity is improved including an improvement in the delivery of the main message.

Response: We thank the reviewer for their positive comments.

The main outcome is that there are competing trends when considering sediment deposition (supply and retention) which produce a nonlinear relationship between net deposition and vegetation density. I think the work could have some potential for influencing restoration projects though in order for that to really happen a wider analysis in terms of general geomorphic settings would be needed. The latter would be outside the scope of the work so I would not ask the authors to address this unless they think some mention of this could serve to enrich the discussion section. The main analyses support the conclusions. The model validation remains a little bit slim and a minor suggestion would be to try finding more data points for validation or trying to replace the grey dashed curve in figure 5 with an ensemble of simulation points. This would improve the quantitative nature of the results. However, this is a suggestion, and the main message of the article could be delivered without implementing this.

Response: First, we agree that the particular optimum in vegetation density would depend on geomorphic setting (e.g., sediment grain sizes, existing topography), but that a consideration of this dependence is beyond the scope of this paper. We also agree that the work could potentially influence restoration decisions. We tried to highlight this potential by connecting the work to the planned sediment diversions aimed at restoring the MRD marshes (Line 31-33). In addition, we have now added a more specific comment in Line 48-50 “For example, a more efficient application of sediment diversion could be achieved by timing sediment input to periods of vegetation density that optimize sediment accretion, which was the focus of this study.”

Second, we agree that having more data for validation would be good. However, we have searched data in the literature, and we could not find another study that provided the range of data needed to make a comparison to the model, which includes measurements sedimentation rate, vegetation coverage, and flow velocity over a cycle of vegetation growth.

Finally, we chose the dashed grey line in Fig. 5 after considering a more complex version of the figure that plotted three modeling curves, corresponding to the three field conditions. That is, each point in Fig. 5 was associated the different optimum curve, because it represented conditions with a different slope (Table 1). The previous version of the figure, which included three modeling curves, is shown below. However, we felt the inclusion of three curves would make the figure too complicated. For clarification, we have added the following statement to the caption. “Note that each measurement occurred under a different water slope (Table 1), which was associated with a different modeled curve of \tilde{q}_a versus vegetation density. The individual curves are shown in the Supplementary S5.” (Line 545). Encouraged by the reviewer’s interests, we have added the following figure to the Supplementary Information Section S5.

Response to Reviewer #3

First, the authors did a fantastic job addressing the reviewer comments, and the manuscript is much improved. I remain enthusiastic about its contribution, and my remaining comments are relatively minor. The most substantive of my comments is that some of what I see as some of the most important advances of the paper, regarding the finding that the exponential deposition profile applies only to special cases and, perhaps, the finding about the importance of water-surface slope, are relatively buried. I would recommend adding a sentence about the exponential deposition profile to the abstract, at a minimum, so that it better reflects the novel contributions of the paper. Likewise, the Discussion focuses primarily on exceptions to the model (i.e., shortcomings, caveats, and discrepancies between the model and observations), rather than highlighting the key contributions of the work and their implications. It seems that at least one opening paragraph that discusses the positive and novel contributions of the model (such as implications of the finding that the exponential deposition model applies only to select cases, or implications of the finding that at low water surface slopes, vegetation does not yield more deposition, and what this means in places where flow has been impounded or otherwise reduced) would improve the organization of this paper. A final organizational comment is that it seems strange to present equations 1-3 in the Results, when this text is really methods.

Response: We thank the reviewer for their positive comments.

First, we agree with that the finding regarding the spatial sedimentation distribution is important. We have added the following sentence to the abstract (Line 20-22) “Two patterns of sedimentation spatial distribution emerge in the simulation, and the exponential distribution only occurs when resuspension is absent. With resuspension, sediment is delivered farther into the marsh and in a uniform distribution”

Second, the suggestion that adding one paragraph to discuss the novel contribution is helpful to improve the organization of the paper. We added the following discussion (Line 183):

“Using a reduced-order representation of flow between a channel and marsh platform, this study explored a wide parameter space, which provided insight into the role of vegetation density on sediment accretion. First, as vegetation density increases, velocity decreases, which is associated with a reduction in shear stress, TKE and diffusivity, all of which favor deposition and sediment retention. However, the reduction in velocity also reduces sediment supply. The competing effects of enhanced retention and reduced supply produce a nonlinear relationship, such that an intermediate vegetation density yields the maximum net deposition. Second, the optimum vegetation density depends on the water surface slope (S), with maximum accretion shifting toward lower vegetation density as S decreases. In particular, for sufficiently small S , such that $\tau_0 \leq \tau_{c,0}$ even for the bare bed, the peak accretion is associated with bare bed ($n=0$), and sediment accretion rate decreases with the addition of vegetation. Finally, vegetation density also affects the spatial distribution of sedimentation. If vegetation density is sufficient to eliminate resuspension, the deposition pattern is exponential, with maximum deposition at the marsh edge (Pattern 1). However, if the vegetation density is not sufficient to eliminate resuspension, particles deposited near the edge can be resuspended and delivered farther into the marsh, resulting in lower accretion at the marsh edge and a more uniform spatial pattern of deposition farther into the marsh (Pattern 2).”

Third, the definitions provided in Eqns. 1-3 are essential information for the reader to easily follow the results and discussion. This is why we think that these equations should appear within the results section, rather than in the methods sections, which is placed later in the paper.

Additional editorial comments:

Lines 14-15: Consider rewriting as “Marsh vegetation, a definitive component of delta ecosystems, ...”

Response: Thanks. We have rewritten the sentence as “Marsh vegetation, a definitive component of delta ecosystems, has a strong effect on sediment retention and land-building, ...”.

Line 47: are  is

Response: Thanks. We have corrected that.

Line 153: Add “of” after “most.”

Response: Thanks. We have added “of” after “most”.

Line 367-369: This seems to be a run-on sentence, and additionally I believe a verb is missing.

Response: Thanks. We see how this sentence confusing. We revised the sentence to “Vegetation coverage, V_c , defined as the fraction of horizontal surface area obstructed by plants, was estimated on April 24, June 3, and July 1, using random quadrat (0.25 m^2) surveys and rectified drone images” (Line 343-345).

Line 373: leave  leaf

Response: Thanks. We have revised “leave” to “leaf”.

Line 376: leave  leaf area

Response: Thanks. We have revised “leave” to “leaf area”. Additionally, we have reviewed the manuscript carefully and revised some other typos.